# New insights into the impact of financial inclusion on economic growth: A global perspective

Mohammad Naim Azimi  *

Faculty of Economics, Kabul University, Kabul, Afghanistan

* naeem.azimi@gmail.com

**Data Availability Statement:** The datasets for GDP growth, credit to the private sector, age dependency ratio, inflation rate, school enrollment rate, population growth rate, and trade openness (total imports of goods and services plus total exports of goods and services) are collected from

## Abstract

Financial inclusion is critical to inclusive growth, proffering policy solutions to eradicate the barriers that exclude individuals from financial markets. This study explores the effects of financial inclusion on economic growth in a global perspective with a large number of panels classified by income and regional levels from 2002–2020. The analysis begins with the development of a comprehensive composite financial inclusion index comprised of penetration, availability, and usage of financial services and the estimation of heterogeneous panel data models augmented with well-known variables. The results obtained from the panel cointegration test support a long-run relationship between economic growth, financial inclusion, and the control variables in the full panel, income-level, and regional-level economies. Furthermore, the study employs a GMM (generalized method of moment) approach using System-GMM estimators to examine the effects of financial inclusion and the control predictors on economic growth. The results of the GMM model clearly indicate that financial inclusion has a significantly positive impact on economic growth across all panels, implying that financial inclusion is an effective tool in fostering rapid economic growth in the world. Finally, the study delves into the causality relationship between the predictors and provides statistical evidence of bidirectional causality between economic growth and financial inclusion, whereas it only supports unidirectional causality relationships from credit to the private sector, foreign direct investment, inflation rate, the rule of law, school enrollment ratio, and trade openness with no feedback causality. Moreover, the study fails to provide causality evidence from the age dependency ratio and population to economic growth.

## 1. Introduction

Financial inclusion is one of the growing research topics that has recently gained popularity in the literature and received considerable attention from scholars, academics, and policymakers alike. However, the theory of financial inclusion emerged during the 1930s (see, *inter alia*, [1]), but the root of a wide-ranging empirical literature dates back to the early 2000s [2, 3]. Besides, since the spark of the millennium development goals by the United Nations, the policy orientation of the financial inclusion nexus with socioeconomic indicators, specifically economic growth, among all others, has gained prominence in the formulation and implementation of

the World Development Indicators that are available at (https://databank.worldbank.org/source/world-development-indicators). Datasets for the indicators of banking penetration, availability, and usage of financial services are collected from IMF's Financial Access Survey available at (https://data.imf.org/?sk=E5DCAB7E-A5CA-4892-A6EA-598B5463A34C). Dataset for the rule of law is collected from the World Bank's Worldwide Governance Indicators available at (https://info.worldbank.org/governance/wgi/). All datasets are publicly available for replications by scholars.

**Funding:** The author received no specific funding for this work.

**Competing interests:** The author has declared that no competing interests exist.

strategies for sustainable development regardless of the social and economic structure of the countries. The multi-dimensionality and non-uniformity of financial inclusion outreach [4] is assumed to foster economic growth through the gradual integration of people into a formal financial system by making financial services affordable and available at a reasonable cost, and, thus, it effects are highly pronounced vis-à-vis other growth drivers [5]. Although financial inclusion is observed as an effective instrument of social inclusion in satisfying the economic desires of poor and financially excluded individuals directly, it combats extreme poverty, reduces income inequality, and encourages human capital creativity indirectly, which, in turn, has a significant impact on the economic growth of a country. In this regard, perhaps, each jurisdiction requires contextual approaches translated into formal policy frameworks to facilitate an effective and extensive outreach of financial inclusion both for included and excluded segments of society to achieve a higher rate of economic growth in the long run [6]. However, financial inclusion-driven growth takes longer than general theoretical expectations, but the development of its policy framework must articulate three key principles: affordability and undue availability of financial services for all segments of society on the supply side; financial literacy and extensive accessibility on the demand side; and consumer protection, accountability, and institutional quality on the governance side [7].

In recognition of the importance of financial inclusion on economic growth, however, most of the recent studies have departed from the foundational theory of financial inclusion and dived either into endogenous or exogenous models. But regardless of the methodology and magnitude of the effects presented, the available empirical literature has taken three main directions. The first group of studies have examined the effects of financial inclusion on economic growth and other socioeconomic predictors in country-specific contexts with limited outcome generalizability (see, for instance, [8–14]). The second group of studies on financial inclusion impact has focused on specific regional classifications, such as Sub-Saharan Africa (SSA), Asia's developing economies, the One Belt One Road Initiative (OBRI), the Organization for Islamic Cooperation (OIC), and the South Asian Association for Regional Cooperation (SAARC) member countries (see, *inter alia*, [15–19]). The third group of empirical studies has focused either on the panels of mixed economies or developed and developing economies (see, for example, [20–25]).

Although an exception is given to empirical studies by Okonkwo and Ifeanyi [26] in low and middle-income countries; Van and Linh [27] in East Asia and the Pacific region; Emara and Mohieldin [28] in the Middle-East and North Africa (MENA); and Ghassibe et al. [29] in the Middle-East and Central Asia, the existing literature reports the non-existence of a comprehensive empirical study to have examined the effects of financial inclusion on economic growth from a global perspective to provide both statistical evidence on the scale of effects and comparative results by region and income-level economies to support extensive policy formulation. Therefore, it is imperative to direct the study by formulating three important questions. First, does financial inclusion have positive effects on economic growth in global, regional, and income-level economies, though some recent empirical studies provide counter-evidence? Second, are the effects of financial inclusion non-monotonic and vary across panels (global, regional, and income-level) due to economic size as represented by GDP growth? Third, are there any causality relationships between financial inclusion and economic growth with a feedback response?

The present study is an attempt to delve into the effects of financial inclusion on economic growth by controlling for major macroeconomic predictors from global, income-level, and regional-level perspectives. As an empirical fact, the literature is still evolving to understand the magnitude, extent, and direction of the effects of financial inclusion on economic growth —an emerging paradox—and the number of studies that have focused on country-specific,

developed, and developing economies with mixed and even confounded results is not sufficient to support a global agenda on the subject. Moreover, the scarcity of a comprehensive empirical study highlighting the effects of financial inclusion on economic growth from a global perspective to facilitate extensive policy comparison is a significant missing gap in the literature and forms the key motivation for the present study.

The remaining sections of the study are structured as follows. Section two presents a review of literature discussing both theoretical and empirical concepts of financial inclusion and growth. Section three presents the data, variables, and construction methodology of the composite financial inclusion index. Section four develops the theoretical model of the study. Section five explains the estimation strategy of the panel data. Section six presents the results and discusses the findings. Section seven concludes the study.

## 2. Literature review

### 2.1 Theoretical background

The existing literature on the finance-growth nexus owes to Schumpeter's [1] initial theory, stating that financial intermediaries are essential to advance technical innovations in businesses to ensure stable economic growth through saving mobilization, project evaluation, risk analysis, transaction, and money circulation conduits [30]. The theory predicts that missed opportunities are caused by inactive assets held both at personal and organizational disposal due to the absence of financial intermediaries to mobilize savings and enlarge money circulation. This, in turn, forces people to rely on wage-based savings and limits money circulation through financially profitable conduits. On the other hand, market imperfection insulates poor citizens from eluding poverty through limitation of access to formal financial products [15], whereas wider access to financial services has been excoriated as one of the most useful tactics for combating poverty, owing to the fact that higher levels of financial inclusion are linked to lower levels of income inequality and higher economic growth [31]. Since then, scholars have attempted to build various theoretical models to capture the notion of growth around the concept of Schumpeter and have provided many definitions to describe financial services (see, *inter alia*, [32–35]). Among all others, Sarma [36] has provided a comprehensive definition for the notion of financial services—that is, "financial inclusion" as a set of formal financial services to bankable individuals and the process through which such services are made available, accessible, and usable at reasonable economic cost. Thus, in line with this definition, financial inclusion can be regarded as one of the key drivers of economic growth through an increase in general consumption, higher profitable investments, a reduction in monetary overhang by the availability of financial services, and a shift from wage-based savings to return on investments [37]. Therefore, it entails two main principles that postulate the theory of financial inclusion. First, the beneficiary theory of financial inclusion, which comprises vulnerable group theory, dissatisfaction theory, and public good theory; and second, the theory of delivery of financial inclusion, comprising public money theory, echelon theory, and private money theory [38]. Thus, the latter—that is, the theory of delivery of financial inclusion—forms the theoretical direction of the present study. However, far from the basic theory of financial inclusion, almost all recent studies have considered either endogenous or exogenous growth models.

### 2.2 Measurement of financial inclusion

Although based on the multi-faceted theory of financial inclusion, there are numerous definitions that have a general consensus on the outreach of financial inclusion to provide excessive financial services to the bankable members of a nation, prioritizing the gradual integration of

the excluded people into the formal financial system of an economy. Thus, giving rise to the conceptual importance, a comprehensive tool is essential to measure the impact of financial inclusion on various socioeconomic indicators. Despite other quasi-mechanisms (see, *inter alia*, [10, 38–40]), this study follows Sarma [41], who enhanced the construction methodology of the composite financial inclusion index using a distance-based approach dissimilar to the human development index adopted by the United Nations Development Programs (UNDP), using average dimension indexes. A three-dimensional approach to construct the composite financial inclusion index is the banking penetration, availability, and usage of financial services, which are defined as follows.

Banking penetration—that is, access to financial services—indicates the number of users of financial services in an economy, measured by the number of deposit accounts per 1,000 adults and the number of depositors per 1,000 adults [41–43]. Here, the first indicator reflects the size of the bankable segment of society, while the second indicator shows the total number of banked individuals, comprising both active and non-active account holders with financial institutions [44, 45]. Next is the availability dimension, which comprises two key indicators, such as the number of banks per 100,000 adults and the number of automated teller machines (ATMs) per 100,000 adults. It reflects the geographical availability of financial services in terms of banking outlets, bank branches, and the ATMs that are available for utilization [40, 46]. Third is the usage dimension, which also comprises two key indicators, such as the number of loan accounts in banks per 1,000 people and the number of borrowers from banks per 1,000 adults. This dimension measures how customers use financial services in the form of transfers, remittances, borrowing, and savings to reflect the efficiency and inclusiveness of the financial services that are available to people in an economy [35, 47].

## 2.3 Review of recent studies

Although the existing literature still evolves in presenting a sufficient number of empirical works on the effects of financial inclusion on various socioeconomic indicators to encourage comprehensive macroeconomic policy attempts (see, for instance, [48, 49]), it owes its first study to Marc et al. [50], who claimed to have found statistical relationships between financial structure and economic growth. For brevity, this section reviews the most recent studies about the effects of financial inclusion on economic growth in different geographical contexts. For instance, Estrada et al. [51] examined the effects of financial systems, banks, and equity markets on economic growth in 125 developing countries. The authors have used simple methods and found that the financial system's outreach postulates significant effects on economic growth, though the results might be confounded due to misspecification and the choice of financial inclusion predictors. Kpodar and Andrianaivo [52] evaluated the effects of information and communication technologies, mobile phone rollout, and the number of deposits per head—that is, the predictors of financial inclusion—on economic growth in a sample of African economies from 1988–2007. The authors employed the system generalized method of moment technique to overcome any endogeneity issue and found that financial inclusion is an effective tool to increase economic growth in the context of Africa.

Masoud and Hardaker [53] employed an endogenous growth model and a set of data for twelve years to examine the effects of stock market development and the banking sector on economic growth in forty-two emerging economies. The authors found that the stock market has a significant influence on the economic growth of the emerging markets and that they move together in the long run. Moreover, they argue that the banking sector is complementary to the stock market in easing customers' access to their desired financial services. Lenka and Sharma [54] examined the effect of financial inclusion (penetration, access, and usage

dimension) on economic growth in India. The authors used a set of time-series data from 1980–2014, a principal component analysis method to construct the financial inclusion index and the autoregressive distributed lag (ARDL) and error-correction methods to estimate the short and long-run effects of financial inclusion on growth. The authors found that financial inclusion has a positive effect on economic growth both in the short and long runs. Moreover, they also provided evidence of a unidirectional relationship between financial inclusion and economic growth.

Le et al. [55] tested the linkage between financial inclusion, growth, and other socioeconomic indicators in 20 Asian economies from 2011–2016 using a random effects model for panel data analysis. The authors found that Asian countries with a higher growth rate have higher financial inclusion to channelize higher economic growth; an inverse association between financial inclusion and unemployment rate; and the role of financial literacy in effectively utilizing the available financial services. Erlando et al. [56] examined the effects of financial inclusion on economic growth using a set of panel data for Eastern Indonesia. The authors employed the modified vector autoregressive method of Toda and Yamamoto, bivariate causality, and dynamic panel vector autoregressive methods. They found a statistically strong nexus between financial inclusion and economic growth, noticing that financial inclusion spurs economic. On the other hand, Nizam et al. [57] analyzed the effects of financial inclusion on economic growth in 63 developed and developing countries over the period from 2014–2017 using threshold regression analysis. The authors found that there is a threshold effect of financial inclusion on economic growth, implying that the effects are positive but are translated at a higher level than in the low level of the financial inclusion index.

Moreover, the existing literature indicates a counter-example about the negative effects of financial inclusion on economic growth by Rodríguez et al. [58]; who analyzed the relationships between them in 71 countries using a set of data spanning from 2007–2016; and applied ordinary least squares, the generalized method of moment with two-way fixed effects, and Granger causality methods to test their developed hypotheses. They found a negative association between financial inclusion and economic growth, highlighting that financial inclusion exerts an adverse effect on growth and a statistically significant causality nexus between them. Meanwhile, Shen et al. [59] used datasets from the WDI (World Development Indicators) and the IMF (International Monetary Funds) to examine the effects of financial inclusion index on economic growth in 105 countries. The authors used spatial data techniques to analyze the relationships between digital financial inclusion, growth, and other control variables and found that digital financial inclusion has a significantly positive impact on the economic growth of the countries.

Finally, Ozturk and Sana [60] examined the effects of financial inclusion on economic growth and environmental quality in forty-two countries linked with the One Belt One Road Initiative (OBRI) using a set of data spanning from 2007–2019. The authors employed pooled ordinary least squares (OLS), two-stage OLS, and the generalized method of moment (GMM) models. They found that financial inclusion has a positive impact on economic growth but has negative effects on environmental quality through the flow of $CO_2$ emissions.

The purview of the existing literature reveals that the empirical studies conducted to examine the effects of financial inclusion on economic growth have left two significant gaps. First, it reports no comprehensive study to reflect the impact of financial inclusion on growth from a global perspective and no comparative results for cross-country groupings by income and regional levels using unified analytical methodology to highlight comprehensive policy implications to support the literal arguments of the paradigm shift—that is, a shift from financial development to financial inclusion as a global agenda. Second, the mixed and confounded results presented by the existing literature have enhanced the paradox of the effects of financial

inclusion as a driver of growth. Therefore, to fill these gaps, it is important to formulate three key hypotheses. $H_1$: As claimed by the initial concept, financial inclusion has a positive impact on economic growth regardless of economic size and structure across the globe. $H_2$: Though the effects are positive on growth, they are non-monotonic and specified by the size of the economies, viz-à-viz, the GDP. $H_3$: While financial inclusion explains economic growth, it is strongly affected by the growth rate of an economy—that is, there is a bidirectional link between them.

## 3. Data and variables

The datasets contain 218 countries across the world, employing annual observations spanning from 2004–2021 compiled from reliable sources and are organized by various panels reflecting income and regional level economies reported by the World Bank classification report [61]. The variables used are consistent with recent empirical literature and include GDP growth, the composite financial inclusion index, school enrollment rate, age dependency ratio, credit to the private sector, the rule of law, inflation rate, trade openness, the Gini index, and population growth rate. Table 1 provides complete information about them. The study employs GDP growth as the dependent variable proxied for economic growth and the composite financial inclusion index as the independent and key variable of interest. However, the construction method of the composite financial inclusion index (CFII) is discussed later; the present study controls for several macroeconomic predictors to avoid any omitted variable bias. Thus, the school enrollment ratio (SER) is used as a proxy for human capital development. Intuitively, an increase in school enrollment increases skills, knowledge, and creativity, thereby stimulating economic growth. Moreover, the age dependency ratio is used to control its effects on growth. Theory predicts that either too young or too old citizens would be cost-burdensome and negatively impact the growth. Credit to the private sector may also influence economic progression, viz-à-viz greater access to credit facilitates higher capital investment, thus spurring economic growth. Studies by Le et al. [55], Sayed and Shusha [43], and Rashdan and Eissa

**Table 1. Description of variables.**

| Full name | Symbol | Description | Sources of compilation |
|---|---|---|---|
| Gross domestic product growth | ECG | It is expressed as an annual percentage and is used as the dependent variable. | WDI |
| Composite financial inclusion index | CFII | CFII is the key variable of interest, and it is constructed using appropriate indicators for banking penetration, the availability of financial services, and the usage of financial services, using the methodology proposed by Sarma [41]. CFII is expressed as positive integers. | FAS |
| School enrollment ratio | SER | This predictor is expressed as a gross percentage of the enrollment of students in secondary education. | WDI |
| Age dependency ratio | ADR | It is the percentage of working age population from 15–65. | WDI |
| Credit to the private sector | CRP | CRP is expressed as a percentage of GDP. | WDI |
| Rule of law | RUL | It is expressed as a percentile rank from 0–100, implying 0 as no rule of law and 100 as a perfect rule of law. | WGI |
| Inflation rate | INF | INF is the official inflation rate and is expressed as an annual percentage. | WDI |
| Trade Openness | TOP | It is the sum of total imports of goods and services to GDP % plus the sum of total exports of goods and services to GDP %. | WDI |
| Gini coefficient | GIN | Gini index is expressed as positive integers ranging from 0–100, where 0 is perfect income equality and 100 is perfect income inequality. | SWIID |
| Population growth rate | PGR | It is expressed as an annual percentage of the population growth. | WDI |

Notes: The Gini index compiled from SWIID is constructed by Solt [63]. WDI = World Development Indicators, FAS = Financial Access Survey, WGI = Worldwide Governance Indicators, SWIID = Standard World Income Inequality Database.

[45] suggest including the inflation rate as a control variable when delving into the effects of CFII on growth. It is important to understand the effects of higher inflationary episodes that cause the saving rates to decrease and suppress the citizens' use of desired financial services. In light of the globalized economy, it is essential to augment the trade openness in the model to measures the cross-country access of financial services by traders. Despite controlling for the income inequality proxied by Gini index, the study also controls for the effects of institutional quality proxied by the rule of law. It is widely documented that the rule of law is an appropriate proxy for institutional quality when analyzing the effects of CFII on growth [62]. Finally, population growth rate is also used to control its effects on economic growth.

Unlike recent empirical studies that followed the construction methodology of the composite index explained by UNDP for HDI (Human Development Index), the present article adopts a more comprehensive method proposed by Sarma [41] to construct the CFII. Literally, financial inclusion outreach is based on three key dimensions, such as banking penetration, the availability, and the usage of financial services by the bankable population. To quantify, each dimension comprises two key indicators, and thereby, each indicator is assigned an appropriate numerical weight (see Table A1 of Appendix A in S1 Appendix). The construction begins with the specification of dimension index observing minimum and maximum integers using $d_i = w_i(A_{ik,t} - m_i / M_i - m_i)$, in which, $d$, $w_i$, $A_i$, $m_i$, and $M_i$ present the normalized integer, weight, actual value of the country $k$ in time $t$, lower limit (fixed by value 0), and upper limit (fixed by 90th percentile rank) of the dimension $i$, respectively [64]. Considering this, the CFII is estimated using the notions of distance ($d$) achievements of points ($d_1$, $d_2$, $d_3$,..., $d_n$) from ($O = 0, 0, 0,..., 0$) being the worst point and from ($W = w_1, w_2, w_3,..., w_n$) being an ideal point as:

$$x_1 = \sqrt{d_1^2 + d_2^2 + d_3^2 + \ldots + d_n^2} \div \sqrt{w_1^2 + w_2^2 + w_3^2 + \ldots + w_n^2} \tag{1}$$

where Eq (1) is used to estimate the normalized Euclidian distance between the achievement point ($x$) and worst position ($O$) on the n$th$ space [65]. Then, to estimate the normalized inverse distance between ($x$) and ideal position ($W$) on the n$th$ space, the following equation is employed:

$$x_2 = 1 - \left( \sqrt{(w_1 - d_1)^2 + \ldots + (w_n - d_n)^2} \div \sqrt{(w_1^2 + \ldots + w_n^2)} \right) \tag{2}$$

Finally, to estimate the CFII, the average of Eqs (1) and (2) are taken as:

$$CFII_i = \frac{1}{2}(x_1 + x_2) \tag{3}$$

This method of CFII construction is widely used in financial econometrics as a standardized predictor of comprehensive financial inclusion index (see, for instance, [43, 46, 66]).

## 4. Model specification

To obtain specification for delving into the effects of composite financial inclusion index on economic growth, this study draws on Kim et al. [15] and Andiansyah [67] and begins with a long-run panel specification as:

$$ECG_{it} = \varphi_i + \eta_{1i}CFII_{it} + \eta_{2i}SER_{it} + \eta_{3i}ADR_{it} + \eta_{4i}CRP_{it} + \eta_{5i}RUL_{it}$$
$$+ \eta_{6i}INF_{it} + \eta_{7i}TOP_{it} + \eta_{8i}GIN_{it} + \eta_{9i}PGR_{it} + u_{it} \tag{4}$$

where $\varphi$ = intercept, $\eta_1 - \eta_9$ = long-run panel coefficients, $t = 1, 2, \ldots, = T$, $i = 1, 2, \ldots, = N$, $u$ = error term of the model following identically and independently normal distribution (*i.i.n.*

*d*.) assumption, and all other variables hold the same meaning as explained before. Except for the rule of law, which is augmented to control for institutional quality on growth, the choice of other explanatory variables is based on recent empirical studies (see, *inter alia*, [27, 67–70]). Moreover, it is expected that the signs of $\eta_1$, $\eta_2$, $\eta_4$, and $\eta_7 > 0$, the signs of $\eta_3$, $\eta_6$, and $\eta_9 < 0$ and the sign of $\eta_5$ to be a-priori indeterminate; that is, based on the advancement of institutional quality in high-income and upper middle-income economies, it may show positive effects on growth, whilst a negative sign is expected in low-income economies due to deteriorating institutional quality.

## 5. Econometric methods

### 5.1 Cointegration test

From the reviewed literature and recent empirical studies on applied panel data techniques, the present study avoids testing the unit root of the predictors, considering the empirical fact that the number of units is greater than the observations in a panel [71–74]. Therefore, it dives into testing for the panel cointegration to establish the long-run nexus amid indicators. For the rejected null of cross-sectional independence in the panel, the use of common panel cointegration tests such as Kao [75] and Pedroni [76] may lead to biased results. Thus, the study employs the panel LM (Lagrange Multiplier) bootstrap cointegration test proposed by Westerlund and Edgerton [77], which allows for cross-sectional dependence and produces accurate results in small samples. It follows the same principal component analysis explained by Westerlund [78] and is initiated as:

$$\hat{S}_{it} = y_{it} - \hat{\delta}_{0i} - \hat{\delta}_{1i}t - \hat{\eta}_i D_{it} - x'_{it}\hat{\beta}_i - (D_{it}x_{it})'\hat{\gamma}_i - \hat{\lambda}'_i \hat{F}_t \tag{5}$$

where $t = 1,\ldots, T$, $i = 1, \ldots, N$ and $x_{it} = x_{it-1} + v_{it}$ exhibiting I(1) series presenting the K-dimensional regressor vector, $D_{it}$ presents the break dummy, $\hat{F}_t$ is the common factor, and $\hat{\lambda}_i$ is the factor loading for principal component analysis. Therefore, with the help of Eq (6) given below, the panel LM test statistics is computed as:

$$\Delta \hat{S}_{it} = \varphi_i + \vartheta_i \hat{S}_{it-1} + \sum\nolimits_{j=1}^{p_i} \vartheta_{ij}\Delta \hat{S}_{it-j} + \varepsilon_{it} \tag{6}$$

where $\varphi$, the change sign $\Delta$, and $\varepsilon$ are the intercept, first difference operator, and the error term of the model, respectively. Using the LM bootstrap, the null hypothesis indicates panel cointegration, given that the bootstrap ($p > 0.05$) for the LM test statistics against its alternative of no cointegration for all cross-sections.

### 5.2 GMM approach

To estimate Eq (4), and considering the properties of the panel data used in this study, whether cointegrated or not, it employs the generalized method of moment (GMM), which is an appropriate econometric approach suitable for the case of this study. The choice of using the GMM technique is based on empirical facts—the existence of cross-sectional dependence among country groupings, panel endogeneity, and the low frequency of observations compared to the number of units in the panel, say, $T < N$ [79]. For brevity, considering a linear regression with endogenous regressors, this study initiates building the GMM model as:

$$y_{it} = x'_{it}\vartheta + u_{it} \tag{7}$$

where $y_{it}$ = the dependent variable, say, economic growth; $u_{it} = N \times 1$ vectors, $\vartheta = K \times 1$ vector for the unknown parameters, $x_{it} = N \times K$ matrix for the explanatory variables, and $u_{it}$ is the

error term of the model. Assuming that there is an endogeneity issue in the panel, it considers a matrix $x_{it}$ with $N \times L$ given that $L > K$, where $z_{it}$ matrix comprises a set of predictors that are strongly correlated with $x_{it}$ but have orthogonality with $u_{it}$, say, they are not highly correlated with the error term. Thus, $z_{it}$ is assumed to be exogeneous following $E(z'_{it} u_{it}) = 0$ assumption [80]. For GMM estimation, there are two approaches, such as System GMM (Sys-GMM) and Difference GMM (Diff-GMM). For the Sys-GMM, the employed equation takes the following form:

$$\ln y_{it} = \varphi \ln y_{it-1} + \vartheta x'_{it-1} + (\eta_i + u_{it}) \qquad (8)$$

where ln = natural log of the predictors, $\varphi$ = coefficient of the lagged dependent variable, and $\vartheta$ = coefficient of the lagged explanatory variables, and other vector predictors hold the same meaning as explained before. Eq (8) is a level equation comprising fixed effects, while Diff-GMM transforms the predictors by first differencing to remove the fixed effects as:

$$\Delta \ln y_{it} = \varphi \Delta \ln y_{it-1} + \beta \Delta x'_{it-1} + \Delta \varepsilon_{it} \qquad (9)$$

where $\Delta u_{it} = \Delta \eta_i + \Delta \varepsilon_{it}$, say, $u_{it} - u_{it-1} = (\eta_i - \eta_i) + (\Delta \varepsilon_{it} - \Delta \varepsilon_{it-1})$, and other variables hold the same meaning as explained before. Now, Eq (9) removes the fixed effects and does not vary along with the time, but the problem of panel endogeneity still remains. Moreover, $\Delta \ln y_{it}$, which is the first-differenced lagged dependent variable is instrumented with its lagged value and its changes are shown by Eq (9). According to Blundell and Bond [80], if the dependent variable is a random walk process, Eq (9) may produce biased and inconsistent estimate of $\varphi$ in finite samples, especially when the time period is short, which is attributed to poor instruments in the model. To that end, preference is given to Sys-GMM estimation as it simultaneously computes two equations. One with a level, and the instruments in first differenced form. Second, with the first difference, the instruments are expressed in the level form. This method includes more moment conditions when the time period is short and the dependent variable is assumed to be a random walk. Therefore, it gains precision and small sample distortion is reduced. The Sys-GMM estimation is based on two approaches, such as 1Sys-GMM (one step system GMM) and 2Sys-GMM (two step system GMM), where the former assumes no heteroskedasticity and serial correlation and the latter corrects them by exploiting a weighting matrix based on the residuals from the 1Sys-GMM.

Moreover, GMM estimation has several empirical advantages over the common techniques for panel data analysis. First, it controls for omitted variable bias, correlation between the variables, and any potential measurement errors. Second, it produces consistent and accurate results of the coefficients for panel samples with N > T. Third, it corrects the unobserved endogeneity by transforming the regressors through differencing and removing the fixed effects. Fourth, in terms of heteroskedasticity and serial correlation, the Sys-GMM, which is an augmentation of the Diff-GMM, is more consistent, robust, and efficient.

As suggested by the existing literature, in this study, the analysis begins with the estimation of Eq (4) by pooled ordinary least squares (OLS) and the least squares dummy variable (LSDV) using fixed effects methods. Doing so leads the study to select an appropriate GMM estimator and avoid misspecification. Therefore, the pooled OLS panel estimate for $\varphi$ is used as an upper bound, while the fixed effect estimates are used as the lower bounds. If the Diff-GMM estimates are close to or below the fixed effects estimates, Sys-GMM is more consistent and efficient as Diff-GMM would be biased due to weak instrumentation. Moreover, it is also important to test for instruments validity, for which Hansen's [81] J-statistics and Sargan's [82] methods are employed to test the validity of the instrumental variables augmented in the GMM model. Rejecting the null hypothesis implies that the instruments are invalid [83], while

failing to reject the null implies otherwise. Furthermore, this study tests the null of no second-order serial correlation in the error term using the Arellano and Bond [79] method. Failing to reject the null implies that no second-order serial correlation exists and that the moment conditions are appropriately specified.

### 5.3 Panel causality test

Finally, this study delves into the causal relationships between economic growth and the composite financial inclusion and employs the proposed model of Dumitrescu and Hurlin [84] causality test for heterogeneous panel. This test is suitable for the panels that exhibit cross-sectional dependence and when T < N, as in this case. Thus, it provides more consistent results than other common methods. The equation used to test the causality between the predictors is expressed as:

$$Y_{it} = \alpha_i + \sum\nolimits_{k=1}^{K} \lambda_i^k Y_{it-k} + \sum\nolimits_{k=1}^{K} \beta_i^k X_{it-k} + \varepsilon_{it} \qquad (10)$$

where $\alpha$, $\lambda$, $\beta$, and $\varepsilon$ are the intercept, the coefficient for the lagged dependent variable, the coefficient for the lagged independent variable, and the error term of the model, respectively. Eq (10) tests the null of no causality between the predictors in the cross-sections, using individual Wald statistics for each unit and averaged Wald statistics for the whole panel (see [84], for technical details).

## 6. Results and discussion

### 6.1 Descriptive statistics

The analysis begins with some important summary statistics about the variables, reported in Table B1 of Appendix B in S2 Appendix. It shows that the mean value for GDP growth is 2.74% for the full panel, while it is 2%, 4.48%, 5.69%, 6.13%, 1.52%, 1.11%, 4.91%, 1.41%, 2.12%, 3.87%, 1.78%, 5.71%, and 3.99% for low-income, middle-income, upper middle-income, high-income, OECD, non-OECD, East Asia and the Pacific, Europe and Central Asia, Latin America and Caribbean, MENA, North America, South Asia, and Sub-Saharan African economies, respectively. On the other hand, the summary statistics indicate that the mean value for the composite financial inclusion index is 0.26, 0.78, 0.73, 0.69, 0.71, 0.82, 0.74, 0.71, 0.73, 0.68, 0.74, 0.79, 0.68, and 0.72 for the full panel, low-income, middle-income, upper middle-income, high-income, OECD, non-OECD, East Asia and the Pacific, Europe and Central Asia, Latin America and Caribbean, MENA, North America, South Asia, and Sub-Saharan African economies, respectively. It reveals that among all others, although the growth rate of the high-income countries has been the highest throughout the period, their CFII rank has relatively been lower than those of the low-income, middle-income, upper middle-income, OECD, MENA, North America, South Asia, and Sub-Saharan African countries. Moreover, another interesting indicator is the rule of law, which shows that its mean value does not necessarily correspond to the growth rate and the mean value of the financial inclusion outreach. For instance, the mean value of the rule of law is 90.33 percentile rank for high-income economies, which is the highest among all others, while its growth rate and CFII average rate are reported otherwise. For brevity, one can read through the variations among the predictors, but the study proceeds to delve into the cointegration among them.

### 6.2 Cointegration analysis

To ascertain the long-run nexus amid predictors, the Westerlund and Edgerton [77] cointegration test by LM bootstraps was computed, and the results are shown in Table 2. For the

**Table 2. Panel cointegration analysis.**

| Test statistics | Full panel | Income level panels | | | | | |
|---|---|---|---|---|---|---|---|
| | | Low-income | Middle-income | Upper middle-income | High-income | OECD | Non-OECD |
| Gt statistics | -5.13*** | -4.91 | -10.18*** | -16.44*** | -9.08*** | -5.16*** | -7.68* |
| | (-6.24) | (-0.88) | (-5.99) | (-14.08) | (-9.31) | (-9.19) | (-2.38) |
| Ga statistics | -13.42*** | -6.66 | -13.42*** | -16.62*** | -11.52*** | -9.36*** | -9.48*** |
| | (-7.03) | (-1.49) | (-7.45) | (-21.95) | (-7.01) | (-8.27) | (-5.44) |
| Pt statistics | -8.99*** | -4.65 | -20.21** | -10.27*** | -11.08*** | -6.93*** | -6.16*** |
| | (-6.05) | (-1.98) | (-3.98) | (-21.43) | (-5.88) | (-6.24) | (-4.88) |
| Pa statistics | -17.11*** | -4.10 | -24.67*** | -13.21*** | -13.48*** | -10.04*** | -8.05*** |
| | (-5.34) | (-1.04) | (-6.29) | (-9.76) | (-7.49) | (-9.59) | (-6.37) |
| Indicators | Regional panels | | | | | | |
| | East Asia and Pacific | Europe and Central Asia | Latin America and Caribbean | MENA | North America | South Asia | Sub-Saharan Africa |
| Gt statistics | -10.41*** | -8.33*** | -4.36*** | -4.89*** | -8.67*** | -3.62*** | -11.89*** |
| | (-4.89) | (-6.13) | (-3.91) | (-6.50) | (-4.00) | (-5.67) | (-3.98) |
| Ga statistics | -13.00*** | -10.28*** | -6.17*** | -7.41*** | -11.04*** | -5.07*** | -16.30*** |
| | (-5.49) | (-3.87) | (-4.99) | (-5.52) | (-3.82) | (-8.17) | (-5.09) |
| Pt statistics | -8.77*** | -10.09*** | -7.75*** | -5.51*** | -9.46*** | -5.19*** | -9.53*** |
| | (-7.21) | (-3.59) | (-8.17) | (-4.39) | (-6.01) | (-4.88) | (-6.03) |
| Pa statistics | -11.99*** | -14.33*** | -10.04*** | -9.37*** | -15.18*** | -8.36*** | -10.04*** |
| | (-7.64) | (-4.18) | (-6.65) | (-5.04) | (-5.49) | (-4.38) | (-6.19) |

Notes:

***, **, and * indicate significance at 1%, 5%, and 10% levels, respectively. () indicates Z-values. The corresponding p-values are robust and are performed under 100 bootstraps replications.

rejected null hypothesis of no cointegration, except for the low-income economies, the findings reveal that there exists significant cointegration among the predictors in all panels. This implies that panel predictors move together in the long run—that is, the composite financial inclusion index, which is the key variable of interest, and other explanatory variables postulate significant effects on economic growth and cannot be deviated from long-run equilibrium. The results are consistent with the findings of Nwanne [85], Hassan [86], Ratnawati [87], and Ain et al. [88], who also established statistical long-run relationships between financial inclusion and economic growth. Moreover, the results satisfy the underlying theory of growth-financial inclusion [14], implying that excessive financial inclusion outreach facilitates higher capital mobility and integration of a higher proportion of unbanked individuals into the formal financial system, which leads to higher economic growth in the long run.

## 6.3 GMM estimates

As for the key results of interest, Tables 3 and 4 report the results of robust GMM estimation—that is, 1Sys-GMM, 2Sys-GMM, and Diff-GMM for the full panel, income level groupings, and regional economies. As discussed earlier, using empirical diagnostics, the Sys-GMM estimators are preferred over the Diff-GMM, and thus, the interpretation of the results and discussion of findings are based on the 2Sys-GMM results, though the results of the 1Sys-GMM are similar to those of the 2Sys-GMM. For robustness, it follows the Windmeijer [89] correction in the standard errors of the 2Sys-GMM estimation to control for the downward biasedness of

**Table 3. Results of the GMM estimates for income level economies.**

| Variables | Full panel | | | Low-income | | | Middle-income | | | Upper middle-income | | |
|---|---|---|---|---|---|---|---|---|---|---|---|---|
| | 1Sys-GMM | 2Sys-GMM | Diff-GMM | 1Sys-GMM | 2Sys-GMM | Diff-GMM | 1Sys-GMM | 2Sys-GMM | Diff-GMM | 1Sys-GMM | 2Sys-GMM | Diff-GMM |
| Composite financial inclusion index | 0.318*** | 0.316*** | 0.322*** | 0.092*** | 0.086*** | 0.114*** | 0.118*** | 0.119*** | 0.123*** | 0.209*** | 0.212*** | 0.205*** |
| | [4.22] | [6.18] | [4.01] | [8.18] | [5.04] | [6.27] | [4.87] | [3.99] | [4.19] | [4.33] | [6.00] | [5.29] |
| Age dependency ratio (%) | -0.024*** | -0.024*** | -0.016*** | -0.318*** | -0.316*** | -0.405*** | -0.299*** | -0.186*** | -0.189*** | -0.202*** | -0.209*** | -0.211*** |
| | [-6.13] | [-5.29] | [-4.68] | [-4.48] | [-4.35] | [-3.39] | [-6.76] | [-5.96] | [-4.62] | [-10.76] | [-9.54] | [-6.47] |
| Credit to the private sector (GDP %) | 0.467*** | 0.433*** | 0.319*** | 0.128*** | 0.130*** | 0.129*** | 0.321*** | 0.322*** | 0.318*** | 0.337*** | 0.332*** | 0.344*** |
| | [10.37] | [8.04] | [5.97] | [5.63] | [4.33] | [7.01] | [6.11] | [4.27] | [3.49] | [7.89] | [11.17] | [6.56] |
| Foreign direct investment (GDP %) | 0.388*** | 0.382*** | 0.397*** | 0.657*** | 0.514*** | 0.640*** | 0.444*** | 0.438*** | 0.420*** | 0.509*** | 0.502*** | 0.510*** |
| | [4.98] | [5.69] | [5.77] | [5.19] | [4.09] | [5.27] | [8.18] | [3.56] | [5.17] | [5.28] | [6.12] | [5.63] |
| Inflation rate (%) | -0.011*** | -0.010*** | -0.102* | -0.437*** | -0.382*** | -0.410*** | -0.113*** | -0.118*** | -0.116*** | -0.215*** | -0.218*** | -0.222*** |
| | [-4.02] | [-5.18] | [-2.19] | [-2.84] | [-3.21] | [-3.36] | [-3.24] | [-4.15] | [-5.92] | [-5.52] | [-3.88] | [-4.37] |
| Rule of Law (percental rank) | 0.423*** | 0.426*** | 0.519** | 0.110*** | 0.112*** | 0.125*** | 0.359*** | 0.348*** | 0.355*** | 0.417*** | 0.417*** | 0.438*** |
| | [11.02] | [9.45] | [2.66] | [6.68] | [10.28] | [9.47] | [6.83] | [4.99] | [4.86] | [6.66] | [6.01] | [5.87] |
| School enrollment ratio (gross %) | 0.101*** | 0.102*** | 0.098*** | 0.403*** | 0.376*** | 0.418*** | 0.287*** | 0.283*** | 0.280*** | 0.309*** | 0.310*** | 0.203*** |
| | [3.85] | [4.03] | [4.44] | [3.87] | [4.94] | [4.17] | [5.22] | [4.89] | [5.05] | [3.00] | [3.03] | [3.88] |
| Trade openness (GDP %) | 2.404*** | 2.398*** | 2.416*** | 1.644*** | 1.562*** | 1.628*** | 1.437*** | 1.429*** | 3.446*** | 3.719*** | 3.716*** | 3.747*** |
| | [8.16] | [7.94] | [5.65] | [4.55] | [4.18] | [6.04] | [11.01] | [10.89] | [10.33] | [4.99] | [4.65] | [5.17] |
| Population growth rate (%) | -0.083*** | -0.081*** | -0.101*** | -1.217*** | -1.018*** | -1.115*** | -0.209*** | -0.210*** | -0.216*** | -0.417*** | -0.412*** | -0.389*** |
| | [-3.69] | [-4.33] | [-3.86] | [-8.19] | [-3.56] | [-4.68] | [-7.05] | [-6.88] | [-5.89] | [-4.88] | [-5.15] | [-6.00] |
| | | | | High-income | | | OECD | | | Non-OECD | | |
| | | | | 1Sys-GMM | 2Sys-GMM | Diff-GMM | 1Sys-GMM | 2Sys-GMM | Diff-GMM | 1Sys-GMM | 2Sys-GMM | Diff-GMM |
| Composite financial inclusion index | | | | 0.432*** | 0.419*** | 0.405*** | 0.248*** | 0.246*** | 0.301*** | 0.114*** | 0.146*** | 0.138*** |
| | | | | [9.32] | [8.44] | [10.10] | [5.15] | [5.09] | [4.98] | [3.87] | [4.18] | [4.21] |
| Age dependency ratio (%) | | | | -0.188*** | -0.189*** | -0.167*** | -0.330*** | -0.318*** | -0.269*** | -0.712*** | -0.714*** | -0.647*** |
| | | | | [-6.03] | [-5.15] | [-5.36] | [-6.45] | [-6.99] | [-5.37] | [-4.98] | [-4.67] | [-3.88] |
| Credit to the private sector (GDP %) | | | | 0.341*** | 0.348*** | 0.267*** | 0.199*** | 0.202*** | 0.147*** | 0.304*** | 0.310*** | 0.286*** |
| | | | | [8.14] | [4.34] | [4.50] | [10.21] | [9.88] | [5.54] | [6.74] | [6.19] | [4.37] |
| Foreign direct investment (GDP %) | | | | 0.610*** | 0.612*** | 0.546*** | 0.445*** | 0.441*** | 0.392*** | 0.601*** | 0.604*** | 0.544*** |
| | | | | [3.84] | [10.11] | [9.22] | [3.84] | [3.96] | [5.02] | [9.17] | [4.99] | [5.66] |
| Inflation rate (%) | | | | -0.119*** | -0.117*** | -0.210*** | -0.099*** | -0.095*** | -0.117*** | -0.240*** | -0.247*** | -0.237*** |
| | | | | [-5.54] | [-6.01] | [-6.12] | [-11.02] | [-10.40] | [-6.49] | [-4.25] | [-4.61] | [-7.02] |
| Rule of Law (percental rank) | | | | 0.537*** | 0.530*** | 0.417*** | 0.647*** | 0.642*** | 0.615*** | 0.237*** | 0.240*** | 0.177*** |
| | | | | [6.77] | [6.18] | [5.34] | [7.35] | [6.41] | [9.03] | [9.01] | [6.77] | [14.32] |
| School enrollment ratio (gross %) | | | | 0.291*** | 0.301*** | 0.288*** | 0.187*** | 0.169*** | 0.154*** | 0.487** | 0.489*** | 0.505*** |
| | | | | [4.05] | [5.44] | [5.18] | [3.66] | [4.39] | [12.67] | [3.66] | [4.17] | [6.35] |
| Trade openness (GDP %) | | | | 4.808*** | 4.810*** | 4.765*** | 4.688*** | 3.689*** | 4.597*** | 2.361*** | 2.359*** | 2.416*** |
| | | | | [6.31] | [5.99] | [6.06] | [4.13] | [5.77] | [4.06] | [4.68] | [4.92] | [6.18] |

(*Continued*)

**Table 3.** (Continued)

| Variables | Full panel | | | Low-income | | | Middle-income | | | Upper middle-income | | |
|---|---|---|---|---|---|---|---|---|---|---|---|---|
| | 1Sys-GMM | 2Sys-GMM | Diff-GMM | 1Sys-GMM | 2Sys-GMM | Diff-GMM | 1Sys-GMM | 2Sys-GMM | Diff-GMM | 1Sys-GMM | 2Sys-GMM | Diff-GMM |
| Population growth rate (%) | | | | -0.189*** | -0.188*** | -0.146*** | -0.205*** | -0.205*** | -0.200*** | -0.110*** | -0.107*** | -0.124*** |
| | | | | [-4.03] | [-4.10] | [-4.29] | [-9.28] | [-10.10] | [-6.78] | [-11.32] | [-10.41] | [-19.18] |
| *Diagnostic checks* | 1Sys-GMM test-statistics | | *p-value* | 2Sys-GMM test-statistics | | *p-value* | Diff-GMM test statistics | | *p-value* | | | |
| Arellano-bond AR (1) | -1.107 | | 0.425 | -0.248 | | 0.735 | -1.39 | | 0.225 | | | |
| Arellano-bond AR (2) | -0.119 | | 0.640 | -0.314 | | 0.667 | -1.78 | | 0.337 | | | |
| Hansen test | 10.33 | | 0.228 | 8.58 | | 0.112 | 5.93 | | 0.520 | | | |

Notes:

***, **, and * indicate significance at 1%, 5%, and 10% levels, respectively. [] indicates test statistics.

standard errors, controlling the instrument matrix, observation weight, the difference-in-Sargan/Hansen test of instrument validity, and the forward orthogonal transform, which is an alternative to the differencing approach of Arellano and Bover [90], preserving sample size in panels with observational gaps.

For simplicity and use of limited space, the results of the pooled OLS and LSDV fixed effects models are omitted from the present study and will be available upon request. Moreover, to simplify reading through the results and to highlight significant findings, the study reports the results by income and regional classifications as shown in Tables 3 and 4.

**6.3.1 Full panel.** The results of the full panel, thereby the world panel, reflect the overall effects of financial inclusion—a key variable of interest—and other explanatory predictors on economic growth, consisting of (218) countries. The results indicate that financial inclusion proxied by CFII (composite financial inclusion index) has a significantly positive impact on the world's economic growth, implying that one percent increase in financial inclusion (penetration, availability, and usage of financial services) increases the world's economic growth by 0.316%, ceteris-paribus. The results are consistent with the theoretical expectations about the positive growth-financial inclusion nexus and empirical findings of Kim et al. [15] for 55 member countries of the OIC (Organization of Islamic Cooperation), Siddik et al. [19] for 24 Asian developing economies, Singh and Stakic [18] for 8 SAARC (South Asian Association for Regional Cooperation) member countries, and Huang et al. [17] for 27 countries of the European Union. Though, for brevity, other explanatory variables, such as age dependency ratio (−), inflation rate (−), foreign direct investment (+), school enrollment ratio (+), trade openness (+), and population growth (−) have their varying expected effects on economic growth, the rule of law, which is augmented in the model to ascertain its mediating effects on growth, shows that institutional quality is significant to ease the impact of financial inclusion on growth. For instance, the results demonstrate that a 1% increase in the percentile rank of the rule of law causes economic growth to increase by 0.321% in a global context. This is supported by recent empirical findings by Valeriani and Peluso [91], Nguyen et al. [68], Salman et al. [12], and Radulović [92], who also documented the effects of institutional quality on economic growth. Furthermore, an economic intuition suggests that higher institutional quality—that is, comprehensive rule of law—facilitates indirect economic growth through various conduits, one of which is financial inclusion outreach.

**Table 4. Results of the GMM estimates for regional economies.**

| Variables | East Asia and Pacific | | | Europe and Central Asia | | | Latin America and Caribbean | | | MENA | | |
|---|---|---|---|---|---|---|---|---|---|---|---|---|
| | 1Sys-GMM | 2Sys-GMM | Diff-GMM | 1Sys-GMM | 2Sys-GMM | Diff-GMM | 1Sys-GMM | 2Sys-GMM | Diff-GMM | 1Sys-GMM | 2Sys-GMM | Diff-GMM |
| Composite financial inclusion index | 0.217*** [6.14] | 0.218*** [6.60] | 0.303*** [4.78] | 0.205*** [7.55] | 0.209*** [6.84] | 0.218*** [8.44] | 0.466*** [10.47] | 0.468*** [7.07] | 0.507*** [6.18] | 0.489*** [3.91] | 0.501*** [4.10] | 0.467*** [3.33] |
| Age dependency ratio (%) | -0.827*** [-9.45] | -0.833*** [-7.06] | -0.815*** [-6.44] | -1.004*** [-3.99] | -1.010*** [-4.15] | -1.008*** [-6.15] | -0.473*** [-5.59] | -0.444*** [-4.89] | -0.394*** [-8.04] | -1.356*** [-3.77] | -1.349*** [-4.18] | -1.111*** [-5.01] |
| Credit to the private sector (GDP %) | 1.372*** [6.12] | 1.378*** [6.29] | 1.405*** [5.88] | 2.012*** [4.18] | 2.019*** [4.22] | 2.229*** [3.89] | 1.017*** [6.90] | 1.120*** [6.48] | 1.142*** [6.57] | 0.917*** [4.14] | 0.920*** [4.27] | 0.858*** [4.99] |
| Foreign direct investment (GDP %) | 0.788*** [6.13] | 0.783*** [5.81] | 0.916*** [6.56] | 1.031*** [4.02] | 1.038*** [4.03] | 1.427*** [4.15] | 0.946*** [3.83] | 0.952*** [3.99] | 1.011*** [3.87] | 1.329*** [3.58] | 1.388*** [4.19] | 1.302*** [4.33] |
| Inflation rate (%) | -1.146** [-2.34] | -1.150*** [-3.69] | -2.019 [-0.65] | -0.747*** [-4.09] | -0.699*** [-4.38] | -1.005*** [-6.33] | -0.349*** [-4.46] | -0.349* [-2.01] | -0.217*** [-5.31] | -1.202*** [-4.66] | -1.207*** [-3.88] | -1.467*** [-7.02] |
| Rule of Law (percental rank) | 0.319*** [4.24] | 0.321*** [6.04] | 0.367*** [4.21] | 0.671*** [5.69] | 0.655*** [3.14] | 0.489*** [3.70] | 0.217*** [5.86] | 0.225*** [5.89] | 0.249*** [5.88] | 0.309*** [6.27] | 0.310*** [3.43] | 0.287*** [6.38] |
| School enrollment ratio (gross %) | 1.428*** [5.44] | 1.032*** [5.38] | 1.025 [0.84] | 1.343*** [5.39] | 1.288*** [5.44] | 1.118*** [5.17] | 0.937*** [3.87] | 0.933*** [3.94] | 0.991*** [3.77] | 1.716** [2.24] | 1.712** [2.99] | 1.448*** [4.11] |
| Trade openness (GDP %) | 1.018*** [8.15] | 1.016*** [6.77] | 1.124*** [5.16] | 2.819*** [9.44] | 2.820*** [8.01] | 2.704*** [8.56] | 2.533*** [3.87] | 2.533*** [4.12] | 2.617*** [4.45] | 1.066*** [10.33] | 1.063*** [10.02] | 1.851*** [8.77] |
| Population growth rate (%) | -0.902*** [-5.49] | -0.900*** [-6.28] | -0.837 [-1.19] | -0.145*** [-6.17] | -0.147*** [-5.55] | -0.244*** [-9.38] | -0.667 [-0.92] | -0.572 [-1.16] | -0.632 [-1.49] | -1.290*** [-3.95] | -1.292*** [-4.18] | -0.888*** [-3.67] |

| Variables | | | | North America | | | South Asia | | | Sub-Saharan Africa | | |
|---|---|---|---|---|---|---|---|---|---|---|---|---|
| | | | | 1Sys-GMM | 2Sys-GMM | Diff-GMM | 1Sys-GMM | 2Sys-GMM | Diff-GMM | 1Sys-GMM | 2Sys-GMM | Diff-GMM |
| Composite financial inclusion index | | | | 0.347*** [9.55] | 0.256*** [8.23] | 0.406*** [6.17] | 0.781*** [4.99] | 0.783*** [4.82] | 0.655*** [5.10] | 0.640*** [3.89] | 0.642*** [4.88] | 0.509 [1.55] |
| Age dependency ratio (%) | | | | -0.668*** [-3.69] | -0.662*** [-4.14] | -0.701*** [-3.99] | -1.817*** [-8.44] | -1.820*** [-5.67] | -1.803*** [-6.22] | -2.136*** [-4.55] | -2.133*** [-4.82] | -2.051*** [-5.06] |
| Credit to the private sector (GDP %) | | | | 0.746*** [4.57] | 0.740*** [4.26] | 0.802*** [4.13] | 0.328*** [4.27] | 0.331*** [4.90] | 0.274*** [4.38] | 1.409*** [6.38] | 1.404*** [5.94] | 1.226*** [6.09] |
| Foreign direct investment (GDP %) | | | | 0.891*** [5.64] | 0.871*** [5.38] | 0.846*** [5.49] | 0.571*** [3.99] | 0.641*** [3.87] | 0.649*** [4.01] | 0.581*** [3.88] | -0.591*** [3.67] | 0.884*** [4.49] |
| Inflation rate (%) | | | | -0.857*** [-4.19] | -0.860*** [-4.25] | -1.118 [-1.03] | -2.301*** [-4.78] | -2.302*** [-4.14] | -2.247*** [-9.33] | -1.002*** [-5.19] | -1.009*** [5.004] | -1.126*** [-6.16] |
| Rule of Law (percental rank) | | | | 0.477*** [6.21] | 0.476*** [3.87] | 0.392*** [3.42] | 0.116*** [3.44] | 0.116*** [6.43] | 0.104*** [6.26] | 0.344*** [6.29] | 0.338*** [6.07] | 0.417*** [3.79] |
| School enrollment ratio (gross %) | | | | 2.001*** [7.03] | 2.007*** [6.57] | 2.110*** [6.81] | 0.366*** [4.57] | 0.372*** [4.59] | 0.409*** [4.33] | 0.845*** [3.78] | 0.844*** [3.92] | 0.913*** [3.58] |
| Trade openness (GDP %) | | | | 1.473*** [4.99] | 1.477*** [4.18] | 1.615*** [7.00] | 1.001*** [3.88] | 1.002*** [4.69] | 0.937*** [4.04] | 1.653*** [5.58] | 1.541*** [4.24] | 1.233*** [8.47] |

(*Continued*)

**Table 4.** (Continued)

| Variables | East Asia and Pacific | | | Europe and Central Asia | | | Latin America and Caribbean | | | MENA | | |
|---|---|---|---|---|---|---|---|---|---|---|---|---|
| | 1Sys-GMM | 2Sys-GMM | Diff-GMM | 1Sys-GMM | 2Sys-GMM | Diff-GMM | 1Sys-GMM | 2Sys-GMM | Diff-GMM | 1Sys-GMM | 2Sys-GMM | Diff-GMM |
| Population growth rate (%) | | | | -0.847*** [-5.62] | -0.849*** [-4.57] | -0.509*** [-4.89] | -1.108*** [-3.37] | -1.009*** [-4.56] | -0.811*** [-8.44] | -0.681*** [-4.99] | -0.772*** [-4.36] | 0.948 [6.00] |
| Diagnostic checks | 1Sys-GMM test-statistics | | *p-value* | 2Sys-GMM test-statistics | | *p-value* | Diff-GMM test statistics | | *p-value* | | | |
| Arellano-bond AR (1) | -0.999 | | 0.630 | -1.091 | | 0.625 | -0.189 | | 0.341 | | | |
| Arellano-bond AR (2) | -1.082 | | 0.114 | -1.118 | | 0.222 | -0.047 | | 0.725 | | | |
| Hansen test | 7.449 | | 0.325 | 9.365 | | 0.325 | 12.117 | | 0.438 | | | |

Notes:

***, **, and * indicate significance at 1%, 5%, and 10% levels, respectively. [] indicates test statistics.

Moreover, the control variables, such as age dependency ratio, inflation rate, and population growth rate, decrease economic growth by 0.024%, 0.010%, and 0.081%, respectively. The negativity of the age dependency ratio implies the reduction of productivity in the world and a declining long-run trend in growth, whilst the negativity of the inflation rate on growth may additionally cause excessive cost-burden for bankable customers and reduce the scope of financial inclusion. Although some recent studies found that population growth has a positive impact on the economy, the current study finds that a 1% increase in population growth rate reduces economic growth by 0.081%. This is consistent with the findings of Easterlin [93], Klasen [11], and Mason and Lee [94] on the combined population projected effects on lowering economic growth by 1 percentage point per year.

**6.3.2 Income-level.**   However, the results are statistically significant for all income-level economies, but they reveal that for low-income countries, financial inclusion has a positive impact and increases economic growth by 0.085%, while comparatively, it increases the economic growth of middle-income, upper middle-income, high-income, OECD, and non-OECD member countries by 0.119%, 0.212%, 0.419%, 0.405%, and 0.146%, respectively. This highlights an important variation in the effects of financial inclusion on economic growth, varying with respect to the income level of the countries. Considering the rule of law as a proxy for institutional quality, the results indicate that its effect also varies across income-level groupings. It shows that, ceteris paribus, one percentile rank increase in the rule of law causes economic growth by 0.112%, 0.348%, 0.417%, 0.537%, 0.642%, 0.240% in low-income, middle-income, upper middle-income, high-income, OECD, and non-OECD countries, respectively. Thus, the variation of the effects of financial inclusion may be due to two key reasons: the economic size of the countries and the implementation of the rule of law in governing, allocating, and using financial resources for the sake of rapid growth. Consistently, Azimi [95] also clearly shows that the non-monotonic effects of the rule of law are based on the varying economic size of the country. The findings indicate that the higher the income level, the greater the impact of financial inclusion on economic growth will be. Furthermore, the results show that the theoretically expected effects of the control variables, such as age dependency ratio (–), credit to the private sector (+), foreign direct investment (+), inflation rate (–), school enrollment rate (+), trade openness (+), and population growth rate (–) on the economic growth of the income-

level grouping are achieved. Not surprisingly, the impact of the control variables on growth is also found to be non-monotonic.

For instance, the credit to the private sector has a significantly positive effects on growth, showing that a 1% increase in credit to the private sector, the economic growth increases by 0.130%, 0.322%, 0.332%, 0.348% in low-income, middle-income, upper middle-income, and high-income economies, respectively. Olowofeso et al. [24], Cuong [23], and Samuel-Hope et al. [22] also found statistical evidence of the positive effects of credit to the private sector on economic growth in various economic contexts. On the other hand, foreign direct investment also posits positive effects on growth. Its effect on growth is 0.514%, 0.438%, 0.502%, 0.612%, 0.441%, and 0.604% for low-income, middle-income, upper middle-income, high-income, OECD (Organization for Economic Cooperation and Development), and non-OECD member countries, respectively. Moreover, the negative association between the inflation rate and the age dependency ratio is lower in high-income but higher in low-income economies. For example, the negative effect of the inflation rate on growth is -0.382% in low-income countries, while it is -0.118%, -0.218%, and -0.117% in middle-income, upper middle-income, and high-income countries, respectively. Studies by Babajide et al. [96]; Lenka and Sharma [97]; Dahiya and Kumar [13]; and Okonkwo and Ifeanyi [26] also provide statistical evidence of the effects of financial inclusion on economic growth in low and middle-income countries. Consistently, Sethi and Acharya [21] found a significant association between growth and financial inclusion in 31 countries across the world, consisting of low, middle, and high-income economies, while Li et al. [20] extended the statistical findings of the effects of financial inclusion in OECD member countries, supporting the findings of the present study.

**6.3.3 Regional-level.** To facilitate better analysis and deeper insights into the effects of financial inclusion on economic growth, the present study delves into the matter using the regional classification of the countries. Comparatively, the results provide much deeper views of the growth-financial inclusion association in regional contexts. The results demonstrate that financial inclusion, which is the key variable of interest, is statistically significant at 1% level and spurs economic growth by 0.218% in East Asia and the Pacific, 0.209% in Europe and Central Asia, 0.408% in Latin America and the Caribbean, 0.501% in MENA, 0.256% in North America, 0.783% in South Asia, and 0.642% in Sub-Saharan African countries. Comparatively, the results indicate that South Asia's growth has the highest reaction to financial inclusion among all others, while Europe and Central Asia's growth rate has the lowest response to financial inclusion. The results are consistent with the findings of Van and Linh [27] in East Asia and the Pacific and Abdul Karim et al. [25] in sixty developed and developing economies, Adalessossi and Kaya [98] and Wokabi and Fatoki [99] in African countries, Thathsarani et al. [2] in eight South Asian countries, Gakpa [100] and Adedokun and Ağa [16] in Sub-Saharan African countries, Emara and Mohieldin [28] in MENA, and Ghassibe et al. [29] in the Middle-East and Central Asia, who also found that financial inclusion is a significant determinant of economic growth and an effective tool to facilitate greater financial integration. From a macroeconomic standpoint, increasing unbanked individuals' access to financial services leads to increased money circulation and credit exchange in the economy, resulting in a significant impact on stable economic growth [101], whereas integrating unbanked individuals into the formal financial system results in a meaningful reduction in tax avoidance, money laundering, and transaction costs [57]. However, the proportional impact of financial inclusion is relatively lower than that of financial deepening, but it continues to encourage more unbanked populations to join the formal financial system to generate higher impacts on economic growth. For instance, in East Asia and the Pacific, the results indicate that financial inclusion significantly increases economic growth by 0.218%, while other predictors, such as credit to the private sector, foreign direct investment, the rule of law, school enrollment rate, and trade openness, also

exert positive effects on growth by 1.378%, 0.783%, 0.321%, 1.032%, and 1.016%, respectively. A quick intra-comparison shows that financial deepening, human capital, and economic openness—that is, credit to the private sector, school enrollment ratio, and trade, respectively—have much higher effects on growth than financial inclusion in East Asia and the Pacific. The same results apply to all regional economies, except for South Asian countries that exhibit a different scale but similar magnitude. It is found that, in South Asia, a 1% increase in financial inclusion significantly causes economic growth to increase by 0.783%. With respect to both the inter-region and intra-region comparisons, the effects of financial inclusion on growth are higher than in other regional and income-level economies. This could be due to obvious factors such as significant advancements and support for financial inclusion in South Asia's three most populous countries, India, Pakistan, and Bangladesh, which are constantly expanding the reach of their financial inclusion services. This finding is also supported by Thathsarani et al. [2] in eight South Asian economies, who found that the comparative effects of financial inclusion are lower than other financial development predictors on growth, and by Park and Mercado [102] in thirty-three developing economies, who provided similar findings on the comparative effects of financial inclusion on economic growth. For the control variables, the results indicate that age dependency ratio, inflation rate, and population growth are statistically significant and posit negative impacts on the economic growth of the regional economies, while credit to the private sector, foreign direct investment, school enrollment rate, and trade openness have positive associations with economic growth. Moreover, the results also show that institutional quality proxied by the rule of law has a significantly positive impact on economic growth, whereas, as theory suggests, higher institutional quality leads to efficient and effective delivery of financial services and thus paves the way for swift economic growth. The results reported in Tables 3 and 4 are statistically robust. The diagnostic checks of the relevant tests are reported at the rear part of the tables.

## 6.4 Causality nexus

Finally, the present study computes the panel causality test of Dumitrescu and Hurlin [84] and reports the results in Table 5, highlighting interesting results. They reveal that there is a significant bidirectional causality relationship between economic growth and financial inclusion at a 1% level in the full panel, income-level panels, and regional panels. Since feedback responses—that is, reverse causality statistics of the control variables—have not been significant, they are not reported in Table 5. The results indicate that except for the age dependency ratio (ADR) and population growth rate (PGR), which are insignificant in causing economic growth in the full and all other classified panels, the rest of the variables, such as credit to the private sector, foreign direct investment, inflation rate, the rule of law, school enrollment ratio, and trade openness, are significant enough to exhibit unidirectional causality to cause economic growth. The results support the findings of Sharma [103], Mlachila et al. [104], Sethi and Acharya [21] in a panel of both developed and developing countries, and Gourène and Mendy [105] in the West African Economic and Monetary Union, who also found directional causality relationships between financial inclusion and economic growth in different economic contexts. The results are in contrast with those of Asmalidar and Pratomo [106], who claimed that there is no causality nexus amid financial inclusion and economic growth.

## 7. Conclusion

The present study explored the effects of financial inclusion on the world's economic growth and extended the analysis to delve into how financial inclusion influences economic growth in a number of panels classified by income-level (low, middle, upper middle, high, OECD, and

**Table 5. Panel causality test results.**

| Direction of causality | Full panel | Income level panels | | | | | |
|---|---|---|---|---|---|---|---|
| | | Low-income | Middle-income | Upper middle-income | High-income | OECD | Non-OECD |
| | W-stat. | W-stat. | W-stat. | W-stat. | W-stat. | W-stat. | W-stat. |
| CFII = >/ ECG | 12.018*** | 6.001*** | 4.877*** | 13.020*** | 6.977*** | 10.045*** | 11.288*** |
| | [0.000] | [0.001] | [0.009] | [0.000] | [0.000] | [0.000] | [0.000] |
| ECG = >/ CFII | 10.485*** | 9.217*** | 7.320*** | 8.090*** | 6.172*** | 5.058*** | 6.257*** |
| | [0.000] | [0.000] | [0.000] | [0.000] | [0.001] | [0.010] | [0.000] |
| ADR = >/ ECG | 0.955 | 1.888 | 0.029 | 1.962 | 0.991 | 1.716 | 1.055 |
| | [0.725] | [0.615] | [0.880] | [0.625] | [0.712] | [0.620] | [0.540] |
| CPR = >/ ECG | 6.433*** | 3.944** | 8.212*** | 6.571*** | 5.673*** | 4.918*** | 4.099*** |
| | [0.000] | [0.038] | [0.000] | [0.000] | [0.002] | [0.007] | [0.009] |
| FDI = >/ ECG | 10.337*** | 3.702** | 7.414*** | 8.027*** | 4.935*** | 1.099 | 4.389*** |
| | [0.000] | [0.025] | [0.000] | [0.000] | [0.008] | [0.570] | [0.009] |
| INF = >/ ECG | 5.942*** | 5.897*** | 9.372*** | 6.332*** | 4.029** | 3.874** | 10.148*** |
| | [0.001] | [0.001] | [0.000] | [0.000] | [0.011] | [0.013] | [0.000] |
| RUL = >/ ECG | 3.617** | 5.133*** | 5.793*** | 4.919*** | 5.638*** | 8.011*** | 11.497*** |
| | [0.044] | [0.004] | [0.002] | [0.006] | [0.002] | [0.000] | [0.000] |
| SER = >/ ECG | 6.448*** | 4.498*** | 7.187*** | 9.462*** | 8.504*** | 6.910*** | 5.099*** |
| | [0.000] | [0.002] | [0.000] | [0.000] | [0.000] | [0.000] | [0.002] |
| TOP = >/ ECG | 5.122*** | 3.668** | 4.872*** | 6.916*** | 5.389*** | 4.745*** | 4.917*** |
| | [0.002] | [0.036] | [0.010] | [0.000] | [0.001] | [0.009] | [0.001] |
| PGR = >/ ECG | 1.440 | 1.077 | 0.817 | 0.225 | 1.014 | 0.839 | 1.217 |
| | [0.620] | [0.740] | [0.825] | [0.850] | [0.650] | [0.733] | [0.640] |
| Indicators | Regional panels | | | | | | |
| | East Asia and Pacific | Europe and Central Asia | Latin America and Caribbean | MENA | North America | South Asia | Sub-Saharan Africa |
| | W-stat. | W-stat. | W-stat. | W-stat. | W-stat. | W-stat. | W-stat. |
| CFII = >/ ECG | 8.467*** | 9.444*** | 5.312*** | 8.098*** | 10.355*** | 6.934*** | 20.119*** |
| | [0.000] | [0.000] | [0.001] | [0.000] | [0.000] | [0.000] | [0.000] |
| ECG = >/ CFII | 3.434** | 8.130*** | 6.282*** | 4.044*** | 5.217*** | 5.467*** | 3.999** |
| | [0.048] | [0.000] | [0.000] | [0.009] | [0.000] | [0.000] | [0.049] |
| ADR = >/ ECG | 0.682 | 1.003 | 0.846 | 0.247 | 0.538 | 0.119 | 0.285 |
| | [0.825] | [0.629] | [0.712] | [0.839] | [0.815] | [0.894] | [0.821] |
| CPR = >/ ECG | 6.819*** | 5.005*** | 9.233*** | 4.902*** | 3.881** | 3.027* | 4.520*** |
| | [0.000] | [0.000] | [0.000] | [0.003] | [0.018] | [0.052] | [0.009] |
| FDI = >/ ECG | 5.901*** | 9.106*** | 6.852*** | 7.001*** | 6.492*** | 8.022*** | 5.816*** |
| | [0.000] | [0.000] | [0.000] | [0.000] | [0.000] | [0.000] | [0.000] |
| INF = >/ ECG | 5.733*** | 2.775* | 5.482*** | 4.088** | 5.156*** | 4.993*** | 9.072*** |
| | [0.000] | [0.068] | [0.000] | [0.039] | [0.000] | [0.008] | [0.000] |
| RUL = >/ ECG | 2.992** | 4.816*** | 5.666*** | 6.387*** | 5.942*** | 5.472*** | 8.110*** |
| | [0.046] | [0.008] | [0.000] | [0.000] | [0.000] | [0.000] | [0.000] |
| SER = >/ ECG | 6.010*** | 4.686*** | 5.333*** | 6.754*** | 4.938*** | 3.445** | 6.879*** |
| | [0.000] | [0.007] | [0.000] | [0.000] | [0.000] | [0.025] | [0.000] |
| TOP = >/ ECG | 5.759*** | 3.715** | 4.647*** | 5.071*** | 5.770*** | 6.350*** | 3.522** |
| | [0.000] | [0.026] | [0.007] | [0.000] | [0.000] | [0.000] | [0.044] |

*(Continued)*

**Table 5.** (Continued)

| Direction of causality | Full panel | Income level panels | | | | | |
|---|---|---|---|---|---|---|---|
| | | Low-income | Middle-income | Upper middle-income | High-income | OECD | Non-OECD |
| | W-stat. | W-stat. | W-stat. | W-stat. | W-stat. | W-stat. | W-stat. |
| PGR = >/ ECG | 1.378 | 1.431 | 1.002 | 1.577 | 0.899 | 0.262 | 0.113 |
| | [0.358] | [0.350] | [0.445] | [0.515] | [0.625] | [0.713] | [0.840] |

Notes:

***, **, and * indicate significance at 1%, 5%, and 10% levels, respectively. = >/ indicates the null of no panel causality relationship. [] indicates *p-values*.

non-OECD) economies and regional-level (East Asia and the Pacific, Europe and Central Asia, Latin America and the Caribbean, MENA, North America, South Asia, and Sub-Saharan Africa) economies using panel datasets spanning from 2002–2020 compiled from the World Bank's World Development Indicators and IMF's Financial Access Survey databases. To test the developed hypotheses, the study employed heterogeneous robust panel cointegration by bootstraps, GMM (generalized method of moment), and heterogeneous panel causality tests for two key objectives: ascertaining the scale of the effects of financial inclusion on economic growth and the causality nexus among them. First, to provide consistent results, the study used the methodology proposed by Sarma [41] and developed a comprehensive composite financial inclusion index comprised of penetration, availability, and usage of financial services for all panels under consideration. Next, the study provides statistically significant cointegration between composite financial inclusion, economic growth proxied by GDP growth, and other control variables, such as age dependency ratio, credit to the private sector, foreign direct investment, inflation rate, the rule of law, school enrollment ratio, trade openness, and population growth rate. The cointegration results support the idea that, in the long run, financial inclusion moves in tandem with economic growth, suggesting a thorough examination of their interactions. To that end, led by the nature of the data, the study employed a GMM approach using 1Sys-GMM, 2Sys-GMM, and Diff-GMM estimators to examine the effects of financial inclusion and other control predictors on economic growth. Based on statistical satisfaction, 2Sys-GMM has attained the preference and, thus, the conclusion is based on the findings from 2Sys-GMM estimators. The results clearly indicate that financial inclusion is a significant variable to influence economic growth in the full panel, income-level panels, and regional-level panels and causes the economies to foster rapid growth. The results obtained from the GMM estimates encouraged the study to delve into the causality relationship between financial inclusion, economic growth, and the control variables. In this regard, the results of the Dumitrescu and Hurlin [84] model extend the findings and provide evidence of bidirectional causality between financial inclusion and economic growth, while the results only support a unidirectional causality running from credit to the private sector, foreign direct investment, inflation rate, rule of law, school enrollment ratio, and trade openness with no feedback response. Moreover, the findings also failed to provide causality evidence for the age dependency ratio-economic growth and population growth-economic growth.

## 7.1 Policy recommendations

The findings extracted from a wide range of sophisticated methods and large panel spectra; suggest three important policy recommendations, among all others. First, it clearly indicates that financial inclusion is an assistive growth conduit in all economies regardless of any

classification, and, thus, it is important to enhance the scope of its coverage via more extensive and swift channels, such as the advancement of financial services through financial technologies. This approach will lead to a higher and quicker integration of the excluded segment of society into formal financial systems and will maximize the effects of financial inclusion on economic growth. Second, it is imperative to gear policies to avert the existing digital gaps stemming from access inequality to digital technology. Informed investments in enhancing digital financial services including mobile banking and digital agent networking will substantially reduce the cost of transactions and encourage more entrants into the financial sector of an economy. Third, both public and private sector organizations need to provide an appropriate platform for hasty adaptation of financial technologies to pave the way for easy use of financial services by customers, enhance financial literacy to support effective use, and extend the scope of financial inclusion coverage. These policy measures will lead to extensive money circulation in the economy and growth channelization via higher financial system integration and investments.

### 7.2 Limitations of the study

Although the results are robust and cannot be doubted, the study suffers from one major limitation—that is, the use of the rule of law as a proxy for institutional quality, which may not be sufficient to capture the effects of institutional quality in its entirety. Future studies may overcome this shortcoming by including all institutional quality predictors, such as control of corruption, voice and accountability, government effectiveness, political stability, and regulatory quality when assessing the effects of financial inclusion on growth.

## Supporting information

**S1 Appendix. Table A1 of Appendix A provides statistical detail about CFII aggregate index.**
(DOCX)

**S2 Appendix. Table B1 of Appendix B provides the descriptive statistics of the panel predictors.**
(DOCX)

## Author Contributions

**Writing – original draft:** Mohammad Naim Azimi.

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
