## [Decision Letter · Decision Letter 0]

25 Oct 2022

PONE-D-22-25867New insights into the impact of financial inclusion on economic growth: A global perspectivePLOS ONE

Dear Dr. Mohammad Naim Azimi,

Thank you for submitting your manuscript to PLOS ONE. After careful consideration, we feel that it has merit but does not fully meet PLOS ONE’s publication criteria as it currently stands. Therefore, we invite you to submit a revised version of the manuscript that addresses the points raised during the review process.

We look forward to receiving your revised manuscript.

Kind regards,

Ricky Chee Jiun Chia

Academic Editor

PLOS ONE

Journal Requirements:

Reviewers' comments:

Reviewer's Responses to Questions

**Comments to the Author**

1. Is the manuscript technically sound, and do the data support the conclusions?

Reviewer #1: Yes

Reviewer #2: Yes

2. Has the statistical analysis been performed appropriately and rigorously? 

Reviewer #1: Yes

Reviewer #2: Yes

3. Have the authors made all data underlying the findings in their manuscript fully available?

Reviewer #1: Yes

Reviewer #2: Yes

4. Is the manuscript presented in an intelligible fashion and written in standard English?

Reviewer #1: Yes

Reviewer #2: Yes

5. Review Comments to the Author

Reviewer #1: The study presents the results of primary scientific research. It analyses effects of financial inclusion on economic growth from global angle with a large number of panels investigated by income and regional aspects from wide time range. The structure of the paper is satisfying. Theoretical background is well written indicating the importance of this research and emphasizing the potential gaps of the recent studies used by authors as basis of their research. Definition of the variables and their measurement are detail described and supported by other studies which additional rises the quality of the defined research model.

Econometric analyses are performed to a high technical standard and outputs described in detail to fully understand the relationships between key exogenous variable and target endogenous variable. The proposed research model is well defined taking into account also relevant control variables (choosed according to significance in prior studies) and so delivering comprehensive view on financial inclusion- economic growth linkage. All conclusion remarks are supported by the data. Some suggestion to author refers to lack of recognition of the research shortcoming and suggestions for future studies. Adding these would rise the quality of the research paper.

Reviewer #2: This paper investigates the impact of financial inclusion on economic growth from a global perspective, using the System-GMM method and classifying full panel, income level, and regional level for analysis, and finally also discusses the causal relationship between the two, which is more comprehensive. However, some of my comments are presented to share with the authors.

1.In my opinion, the author should write clearly the purpose and significance or importance of the research in the abstract section.

2.The introduction section feels repetitive with the literature section, mostly a list of literature. In the introduction section, the authors propose two key motives, "reflect consistent results on the effects of financial inclusion" (page 2) and " emerging paradigm shift" (page 2), which in my opinion means the same thing, and I do not understand the so-called two key motives of the authors.

3.Personally, I think there are many articles that do this direction, and I suggest that the authors provide a more novel viewpoint or perspective.

4.The literature review is preferably problem-oriented. What is the problem of the study? It is more laborious for the reader to read, and most literature covers too many issues and is not focused.

5.In the literature review section, the authors mention unemployment, stocks, poverty reduction, and the environment, etc., which I feel is far from the topic of this paper, so I suggest reducing the description of such cases. For example, "has negative effects on environmental quality through the flow of CO2 emissions" can be left out, which does not relate to the topic of the paper.

6.The Hausman test can determine whether a fixed effects model is chosen.

7.Whether Diff-GMM or Sys-GMM, the implementation of the GMM method actually requires preconditions. the GMM as an extension of instrumental variables does have natural advantages in controlling for endogeneity, but a set of assumptions should be satisfied, and it is suggested that the authors add robustness tests to ensure the validity of the estimation results. Examples include the Overidentification test or Hansen's J Test, testing Subsets of Orthogonality condition, classification based on different criteria, substituting with other similar variables, etc.

8.The premise of using System-GMM is that the data should be near the steady state, which means that the samples or individuals cannot be too far from the steady state during the observation period, otherwise the changes of these variables will be more related to the fixed effects, please confirm again whether the condition is satisfied.

9.Please report your options: vertical departure or difference; instrumental variables; lags of several periods; what kind of robust to choose (nonrobust, cluster-robust, or Windmeijer-corrected cluster-robust errors).

10.“considering the properties of the panel data used in this study, whether cointegrated or not, it employs the generalized method of moment (GMM), which is an appropriate econometric approach suitable for the case of this study”(page 7).In that case, is cointegration still needed in this paper? What is the significance of co-integration?

11.Please reconfirm whether the figures in the table are all three stars, i.e., significant at 1% condition. Please provide the test statistics in Table 6-Panel causality test results.

12.The author needs to adjust the format, or at least update the page and line numbers.

13.In the conclusion section, it is recommended to further discuss in depth what policies should be proposed in response to the findings of this paper. The policies proposed by the authors "First, governments need to turn substantial focus on the advancement of infrastructure ......"(page 15) are all ways in which financial inclusion should be achieved. The policy measures are unclear and unimpressive. Please suggest some specific and relevant policy measures from a global perspective based on the findings of the study.

14.Please identify and critically articulate the limitations of the study.

6. PLOS authors have the option to publish the peer review history of their article (what does this mean?). If published, this will include your full peer review and any attached files.

Reviewer #1: No

Reviewer #2: No

---

## [Author Response · Author response to Decision Letter 0]

28 Oct 2022

Response to reviewers 

Manuscript: New insights into the impact of financial inclusion on economic growth: A global perspective

Date: October 28, 2022 

Acknowledgement

First of all, I thank the respected editor and honorable reviewers for their constructive and valuable comments. Indeed, the incorporation of these comments has significantly improved the quality of the paper. All comments are fully incorporated into the manuscript. The respected reviewers may find the point-to-point revisions in the manuscript through the assigned colors. For reviewer #1, green; for reviewer # 2, yellow; for similar comments from reviewers #1 and # 2, grey; and for the editor’s comments, light blue, have been used. Furthermore, the revision can also be tracked via the track changes applied to the original manuscript.

Response to reviewer #1

Some suggestion to author refers to lack of recognition of the research shortcoming and suggestions for future studies. Adding these would rise the quality of the research paper.

Response: Sir, Thanks a lot for this valuable comment. The study imitations have been added in section 7 coded as “7.2 Limitations of the study.”

Response to reviewer #2 

1. In my opinion, the author should write clearly the purpose and significance or importance of the research in the abstract section.

Response: Very true sir. The abstract has been motivated by the incorporation of the importance of the study.

2. The introduction section feels repetitive with the literature section, mostly a list of literature. In the introduction section, the authors propose two key motives, "reflect consistent results on the effects of financial inclusion" (page 2) and " emerging paradigm shift" (page 2), which in my opinion means the same thing, and I do not understand the so-called two key motives of the authors.

Response: Thank you sir for this valuable comment. To incorporate this comment, the following actions have been taken:

1) The repetition of sources in the introduction and literature review parts has been minimized to only two reference that provides the base of the notion and definition. The rest are revised and updated.

2) Based on your valuable comment, page 2, paragraph 4 has been totally revised to make it clearer. Now the key motivation is emphasized and the revised paragraph (4) clearly conveys the message.

3. Personally, I think there are many articles that do this direction, and I suggest that the authors provide a more novel viewpoint or perspective.

Response: Thank you sir for this constructive comment. From two aspects, we have gone through this direction. First, the journal’s “author guide” and instructs to follow this direction; and second, the acceptable standards. The novelty of the paper is clearly emphasized and we believe that this paper is rare in the existing literature.

4. The literature review is preferably problem-oriented. What is the problem of the study? It is more laborious for the reader to read, and most literature covers too many issues and is not focused.

Response: Thank you so much sire for this comment. The problem of study has been clearly indicated in the last paragraph of the section, page 5 “Literature review”. I think that is fair enough to reflect on the problems and missing gaps in the literature. If you are still not convinced, please let us know to highlight it more. 

5. In the literature review section, the authors mention unemployment, stocks, poverty reduction, and the environment, etc., which I feel is far from the topic of this paper, so I suggest reducing the description of such cases. For example, "has negative effects on environmental quality through the flow of CO2 emissions" can be left out, which does not relate to the topic of the paper.

Response: Very true sir. The literature review has been revised and the irrelevant subjects have been removed.

6. The Hausman test can determine whether a fixed effects model is chosen.

Response: Thank you, sir. As an empirical approach to choose between Diff-GMM and Sys-GMM, the paper estimated the pooled OLS and fixed effects models. Therefore, the Hausman test has not applied to choose between fixed or random effect models.

7. Whether Diff-GMM or Sys-GMM, the implementation of the GMM method actually requires preconditions. the GMM as an extension of instrumental variables does have natural advantages in controlling for endogeneity, but a set of assumptions should be satisfied, and it is suggested that the authors add robustness tests to ensure the validity of the estimation results. Examples include the Overidentification test or Hansen's J Test, testing Subsets of Orthogonality condition, classification based on different criteria, substituting with other similar variables, etc.

Response: Thanks a lot, sir. Despite having all sets of econometric assumptions satisfied for the estimation of GMM, the diagnostic tests such as Hansen, Arellano-bond AR(1) and AR(2) are also reported in the rear part of Tables 4 and 5.

8. The premise of using System-GMM is that the data should be near the steady state, which means that the samples or individuals cannot be too far from the steady state during the observation period, otherwise the changes of these variables will be more related to the fixed effects, please confirm again whether the condition is satisfied.

Response: Exactly, sir. The near steady state of the panels classified under income-level and regional level has paved this way for the affirmation of the use of Sys-GMM over the Diff-GMM. However, the comparative results of the POLS and FE models also favor the use of the Sys-GMM model.

9. Please report your options: vertical departure or difference; instrumental variables; lags of several periods; what kind of robust to choose (nonrobust, cluster-robust, or Windmeijer-corrected cluster-robust errors).

Response: Thank you so much for this comment. It is now clearly reported in the manuscript (page 10, last paragraph) that for robustness, the study follows the Windmeijer (2005), which is an alternative to the differencing approach of Arellano and Bover (1995).

10. “considering the properties of the panel data used in this study, whether cointegrated or not, it employs the generalized method of moment (GMM), which is an appropriate econometric approach suitable for the case of this study” (page 7). In that case, is cointegration still needed in this paper? What is the significance of co-integration?

Response: Thank you, sir, for this constructive comment. In the case of our study, in an empirical sense, there is no need to estimate cointegration tests, while authors may still apply cointegration to examine any long-run relationship between the variables. Since the study hypothesized the existence of a long-run nexus amid predictors, it is beneficial to provide statistical evidence rather than making irrelevant judgments. 

11. Please reconfirm whether the figures in the table are all three stars, i.e., significant at 1% condition. Please provide the test statistics in Table 6-Panel causality test results.

Response: Thanks a lot, sir. The significant levels are different and all are not significant at a 1% level. The W-statistics were already reported in the table but the p-values were omitted. Now, the p-values are reported under each W-stat. and thus, the relevant stars, *** presents 1%, ** presents 5%, and * presents 10% significance.

12. The author needs to adjust the format, or at least update the page and line numbers.

Response: Sir, thank you so much. The paper has been reformatted according to the journal’s requirements and the comment of the editor.

13. In the conclusion section, it is recommended to further discuss in depth what policies should be proposed in response to the findings of this paper. The policies proposed by the authors "First, governments need to turn substantial focus on the advancement of infrastructure ......"(page 15) are all ways in which financial inclusion should be achieved. The policy measures are unclear and unimpressive. Please suggest some specific and relevant policy measures from a global perspective based on the findings of the study.

Response: Very true sir. Two actions have been taken for this comment.

1) The recommendation part has been separated and are placed under “7.1 Policy recommendations” in section 7.

2) Policy measures are updated and connected to the findings to reflect clear measures from a global perspective.

14. Please identify and critically articulate the limitations of the study.

Response: Sir, I thank you so much for this constructive comment. The limitations of the study have been added in section 7 “7.1 Limitations of the study.”

Response to the editor

I thank a lot the respected editor for addressing the issues in the paper. For incorporating your comments, the following actions are taken:

1) The format of the paper is adjusted according to the journal’s requirement.

2) The referencing style has been changed to numbering style as required by the journal.

3) Title page has been added in the manuscript as required by the journal.

4) Main headings, sub-headings, and sub-sub-headings have been revised according to the journal’s instruction.

---

## [Editor Report · Decision Letter 1]

3 Nov 2022

New insights into the impact of financial inclusion on economic growth: A global perspective

PONE-D-22-25867R1

Dear Dr. Mohammad Naim Azimi,

We’re pleased to inform you that your manuscript has been judged scientifically suitable for publication and will be formally accepted for publication once it meets all outstanding technical requirements.

Kind regards,

Ricky Chee Jiun Chia

Academic Editor

PLOS ONE
---

## [Editor Report · Acceptance letter]

8 Nov 2022

PONE-D-22-25867R1 

New insights into the impact of financial inclusion on economic growth: A global perspective 

Dear Dr. Azimi:

I'm pleased to inform you that your manuscript has been deemed suitable for publication in PLOS ONE. Congratulations! Your manuscript is now with our production department. 

Kind regards, 

on behalf of

Dr. Ricky Chee Jiun Chia 

Academic Editor

PLOS ONE